

# A simulated observation database to assess the impact of IASI-NG hyperspectral infrared sounder

Javier Andrey-Andrés[1], Nadia Fourrié[1], Vincent Guidard[1], Raymond Armante[2], Pascal Brunel[3], Cyril Crevoisier[2], and Bernard Tournier[4]

[1]CNRM, Météo France and CNRS, 42 Av. Gaspard Coriolis, 31057 Toulouse, France
[2]Laboratoire de Météorologie Dynamique, IPSL, CNRS, Ecole Polytechnique, Palaiseau, France
[3]Centre de Météorologie Satellitaire, Météo France, Av. de Lorraine, 22037 Lannion, France
[4]Noveltis, Labège , France, currently working at Spascia, Toulouse, France

*Correspondence to:* Nadia Fourrié (nadia.fourrie@meteo.fr)

**Abstract.** The highly accurate measurements of the hyperspectral Infrared Atmospheric Sounding Interferometer (IASI) are used in Numerical Weather Prediction (NWP), atmospheric chemistry and climate monitoring. As the second generation of the European Polar System (EPS-SG) is being developed, a new generation of IASI instruments has been designed to fly on board the MetOp-SG constellation: IASI New Generation (IASI-NG). In order to prepare the arrival of this new instrument, and to evaluate its impact on NWP and atmospheric chemistry applications, a set of IASI and IASI-NG simulated data was built and made available to the public to set a common framework for future impact studies. This paper describes the information available in this database and the procedure followed to run the IASI and IASI-NG simulations. These simulated data were evaluated by comparing IASI-NG to IASI observations. The result is also presented here. Additionally, preliminary impact studies of the benefit of IASI-NG compared to IASI on the retrieval of temperature and humidity in a NWP framework are also shown in the present work. With a channel dataset located in the same wave numbers for both instruments, we showed an improvement of the temperature retrievals along all the atmosphere with a maximum in the troposphere with IASI-NG and a lower benefit for the tropospheric humidity.

## 1 Introduction

A huge quantity of improvements have taken place in the first decade of the 21st century with the launch of new infrared sounders such as the Atmospheric InfraRed Sounder (AIRS) in 2002 (Aumann et al., 2003), the Infrared Atmospheric Sounding Interferometer (IASI) in 2006 (Cayla, 2001; Chalon et al., 2001) and the Cross-track Infrared Sounder (CrIS) in 2011 (Glumb et al., 2003). These instruments have drastically raised the amount of information available for meteorological purposes compared to the precedent HIRS (High-resolution Infrared Radiation Sounder) infrared sounder, launched from the late 70's onwards, which offers 19 infrared and 1 visible channels against the thousands of channels available in this new generation of instruments.

The first of these advanced infrared sounders, AIRS, was launched in 2002 with experimental purposes. It is a grating spectrometer providing 2378 channels with approximately 1 cm$^{-1}$ spectral range resolution covering the range from 3 to





15 $\mu$m. For computational cost reasons as well as the fact that many of the information are redundant, this huge amount of channels cannot be assimilated in Numerical Weather Prediction (NWP) models. Hence, further studies were carried out to select appropriate channels (Rodgers, 1996; Susskind et al., 2003). Despite of its experimental conception, AIRS was soon assimilated by operational meteorological models. First attempts of using AIRS radiances lead to an improvement around

0.5-1% in the NWP index use by the MetOffice to quantify the accuracy of NWP models (Collard et al., 2003). Although this improvement was relatively small, it was encouraging as it was obtained assuming a conservative approach. Further studies found that the addition of AIRS data to the observation system improved long-range forecasts (Le Marshall et al., 2006). Fourrié and Thépaut (2003) also demonstrated the usefullness of channel selection.

IASI is the second advanced infrared sounder launched in the last decade (2006). IASI is an infrared Fourier transform

interferometer and is the first instrument of this type to fly as a part of an operational meteorological mission. IASI registers the IR spectrum between 3.6 and 15.5 $\mu$m, providing 8461 channels with a spectral apodized resolution of 0.5 cm$^{-1}$ at a spectral sampling of 0.25 cm$^{-1}$. As in the case of AIRS, preliminary studies showed that some channels were unsuited for an assimilation because of the computing costs and of the existence of correlated information between contiguous channels (Rabier et al., 2002). Preliminary studies about the assimilation of IASI data in the ECMWF (European Centre for Medium

Weather Forecast) model found a mainly positive impact for IASI assimilation and a better quality of the measurements in the 15 $\mu$m $CO_2$ band compared to AIRS (Collard and McNally, 2009). The first Met Office tests about the assimilation of IASI radiances showed a positive impact in the global model which already assimilated data from other sounders (Hilton et al., 2009). Guidard et al. (2011) studied the impact of IASI assimilation in both global and regional Météo-France models. They showed that a quite good impact on the forecast skills for both large-scale variables and precipitation events was found in extra-tropical

regions by the global model ARPEGE (Courtier et al., 1991), and by the AROME regional model (Seity et al., 2011). Hilton et al. (2012), in a review of the main IASI results after 5 years of IASI in operations, remark that the impact scores in global models have been particularly impressive even though IASI is assimilated into an analysis system that is already very well characterized with around 10 microwave and 5 additional infrared sounders, in addition to conventional in-situ measurements.

Although the first objective of atmospheric sounders was to get the temperature and humidity profiles for meteorological

applications, this new generation instruments made possible invaluable advances in the atmospheric chemistry field. Thus, many advances were made in the measurement of trace gases and aerosols (Clerbaux et al., 2009, and references therein) which are key parameters for environmental and climate variables. IASI evidenced a potential good impact of $O_3$ and CO on air quality forecasts and carries on the long-term chemical records started with other instruments (Dufour et al., 2012). Furthermore, some reactive species thought to be undetectable from space such as ammonia have also been measured, thanks

to the excellent signal-to-noise IASI ratio (Clarisse et al., 2009). Other major atmospheric events like volcano eruptions, desert dust intrusions, fires or pollution events can be monitored using IR sounders data (Karagulian et al., 2010; Klüser et al., 2013; Capelle et al., 2014; Clerbaux et al., 2009; George et al., 2009).

This race to provide more effective and accurate instruments still goes on with a new generation of sounders, such as IASI-NG (Bermudo et al., 2014; Crevoisier et al., 2014) on board theMetOp Second generation (MetOp-SG) and IRS (Infrared

sounder) on board the MeteoSat Third Generation satellites, which will be launched from 2020. The former of these two





instruments figures over the same spectral range as IASI with a noise reduction of at least a factor two and a twice higher spectral resolution. IRS will fly on a geostationary satellite providing IR spectrum measurements over Europe every 30 minutes.

Observing Simulated System Experiments (OSSE) are commonly conducted to assess the impact of future observing system on the description of the state of the atmosphere for meteorological purposes (Atlas, 1997; Masutani et al., 2010) or air quality

(Timmermans et al., 2015). During these experiments, the full observation database is built from a realistic description of the state of the atmosphere, the Nature Run. The content of the database is fed into NWP models or in chemistry algorithms whose results are evaluated against a reference set. OSSE are very useful to estimate future observing system impact but at a high computation cost because OSSE mimic the state of the art data assimilation systems.

In this work we built a database of IASI/IASI-NG radiances to evaluate the impact of IASI-NG data with respect to IASI

data using two radiative transfer models: 4A (Scott and Chedin, 1981; Tournier et al., 1995; Chaumat et al., 2012) and RT-TOV (Matricardi et al., 2004; Hocking et al., 2015). Having two sets of simulations enables to carry out observation impact experiments using a different RT model than the one linked to the simulation algorithm, i.e. if the user employs RTTOV in the retrieval algorithm, 4A simulation dataset could be used. This database is then employed to evaluate the impact of IASI-NG with respect to IASI on the retrieval of the temperature and humidity profiles.

First, we present here a common database of IASI and IASI-NG simulations to assess the expected benefits of the latter instrument. This database will serve as common base for future impact studies on the impact of IASI-NG. A preliminary impact study in a 1D-Var retrieval context is presented in the last sections of this work. An exhaustive description of the atmospheric state has been built in the middle of four different dates for each year's season to serve as base for future evaluation of IASI-NG impact. This paper is organized as follows: Section 2 describes in detail the IASI-NG sounder and the main differences with

the IASI sounder. Sections 3 and 4 depict the procedure devised to build the simulated observation database and the main results about the information included in the latter. Sections 5 and 6 deal with the methodology used for the evaluation of using IASI-NG for the retrieval of temperature and water vapour profiles. Finally, the summary and conclusions are provided in section 7.

## 2 IASI and IASI-NG instruments

IASI (Cayla, 2001; Simeoni et al., 1997) is a space-borne interferometer able to characterize the Earth infrared spectra in the range from 645 to 2760 cm$^{-1}$ (15.5 to 3.63 $\mu$m) with a spectral resolution of 0.5 cm$^{-1}$, a spectral sampling of 0.25 cm$^{-1}$ and 8461 channels divided in three different bands:

- Band 1, from channel 1 to 1997 (645.00 to 1144.00 cm$^{-1}$, 15.50 to 8.74 $\mu$m), used for temperature, surface properties, clouds, carbon dioxide and ozone retrievals.

- Band 2, from channel 1998 to 5116 (1144.25 to 1923.75 cm$^{-1}$, 8.74 to 5.20 $\mu$m), used for the retrieval of water vapour, methane and nitrous oxide.

- Band 3, from channel 5117 to 8461 (1924.00 to 2460.00 cm$^{-1}$, 5.20 to 3.62 $\mu$m), sensitive to temperature, surface properties, carbon monoxide, carbon dioxide and nitrous oxide. It is not used at Météo-France because of its larger noise.





The pixel size at nadir viewing is 12 km, being acquired by 4 pixels per field of view. Each scan line is compounded by 30 views. The measurement accuracy is expected to be better than 1 K for temperature retrievals and 10% below 500 hPa for relative humidity retrievals with a vertical resolution finer than 1 km (Diebel et al., 1996). Currently, two IASIs are in flight

on board MetOp-A and MetOp-B satellites, launched in 2006 and 2012 respectively. A third IASI will be launched in 2018 on board MetOp-C satellite. In total, the IASI program is expected to provide a minimum of 15 years of data with good radiometric and spectral stability.

IASI-NG, the next generation of the IASI instrument will be on board EUMETSAT MetOp Second Generation satellites (MetOp SG). The first of MetOp SG satellites is expected to be launched in 2021 and will have an overpassing frequency of

twice a day in mid-latitudes like IASI. To achieve a global coverage, the instrument will perform 14 views per line along the satellite track. This corresponds to 2×7 mirror positions covering a swath of about 2×1000 km (Bermudo et al., 2014) along the satellite track. Each instrument single field-of-view is composed by an array of 4×4 pixels with a size of 12 km at nadir. The surface covered by each view corresponds to a square of 100 km of side length.

The IASI-NG instrument concept is based on a Mertz Interferometer allowing to assess the Self Apodization issue by a

Field Effect Compensation. The Field Effect Compensation allows to correct the differences in the optical path of the different rays that build the interferogram and requires to introduce additional optical elements (Baker, 1980). In the case of the Mertz interferometer this correction of the optical path is performed by two prisms with an adequate refractive index. The noise requirements for IASI-NG are to divide the IASI noise by a factor of two. The IASI-NG noise is related to the optical properties of the prism material. At the current instrument development stage, two different materials are considered viz KBr and ZnSe.

The former of the two materials has been chosen because of its better spectral response, specially at the beginning of the measured spectral range. The noises of IASI and IASI-NG considering both materials are shown in Figure 1. Although KBr noise signature is slightly higher than that of ZnSe, the latter material does not meet the requirements at the beginning of the first band with expected noise values 50% higher than IASI-NG specifications. For the time being, ZnSe material has been discarded for the final instrument configuration. Future studies using the database presented in this work, may help to decide

which of the two materials should be used for the final instrument. The IASI-NG noise reduction is achieved by using twice the integration time of IASI, and the increment of the instrument spectral resolution is accomplished thanks to the 8 cm optical path of IASI-NG, twice the 4 cm IASI optical path.

IASI-NG will have an additional band compared to IASI passing from 3 to 4 bands. The IASI band 3 will be divided into two different bands whereas the limits for the first two bands are quite similar to IASI ones. The limits for the four IASI-NG bands

are: B1 from 645 to 1150 $cm^{-1}$, B2 from 1150 to 1950 $cm^{-1}$, B3 from 1950 to 2300 $cm^{-1}$ and B4 from 2300 to 2760 $cm^{-1}$.

Table 1 compares the main features of IASI and IASI-NG instruments. The consequences of this path increment on the Instrument Response Function (ISRF) is shown in Figure 2, where IASI-NG ISRF width is approximately half of that of IASI. Consequently, for IASI-NG, the spectral sampling is made each 0.125 $cm^{-1}$ instead of 0.25 $cm^{-1}$ for IASI. IASI-NG will have 16,921 channels from 645 $cm^{-1}$ to 2760 $cm^{-1}$.





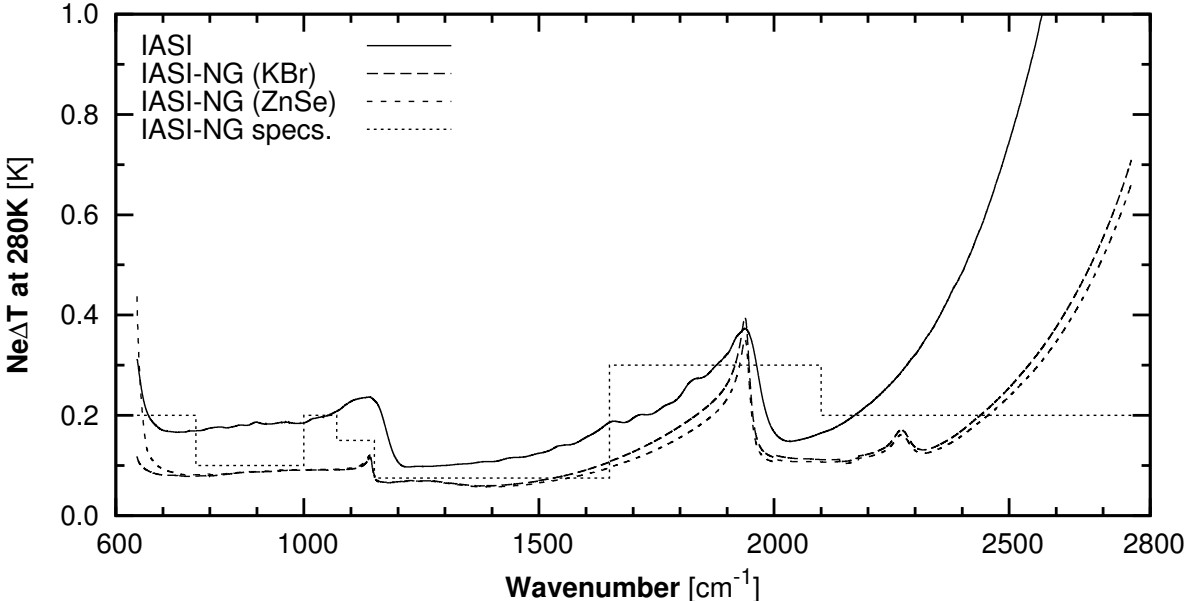

**Figure 1.** NedT noise at 280 K for IASI and IASI-NG (source data from E. Pequignot, CNES)

| Main features | IASI | IASI-NG |
|---|:---:|:---:|
| Pixels in field of view | 4 | 16 |
| Channels | 8461 | 16921 |
| Radiometric resolution (NedT) | | IASI/2 |
| Spectral sampling | 0.25 cm$^{-1}$ @L1C | IASI/2 |
| Abs. Radiometric Calibration | <0.25 K @280K | IASI/2 |
| Spectral bands | 3 | 4 |

**Table 1.** main features of IASI and IASI-NG sounders

## 3   Database construction

To build the database consisted, first of all, in getting the most accurate description possible of the state of the atmosphere, so that data could be used for the simulations of both IASI and IASI-NG measurements. Four different dates were chosen for the year 2013. Each date falls in the middle of each season: February 4$^{th}$ (North Hemisphere Winter), May 6$^{th}$ (NH Spring), August 6$^{th}$ (NH Summer) and November 4$^{th}$ (NH Autumn), to cover the maximum of possible meteorological variability. These 4 periods of 24 hours begin at 21:00 UTC on the previous day. Once these data have been compiled and formatted, they were used to feed two radiative transfer (RT) models to simulate measurements.



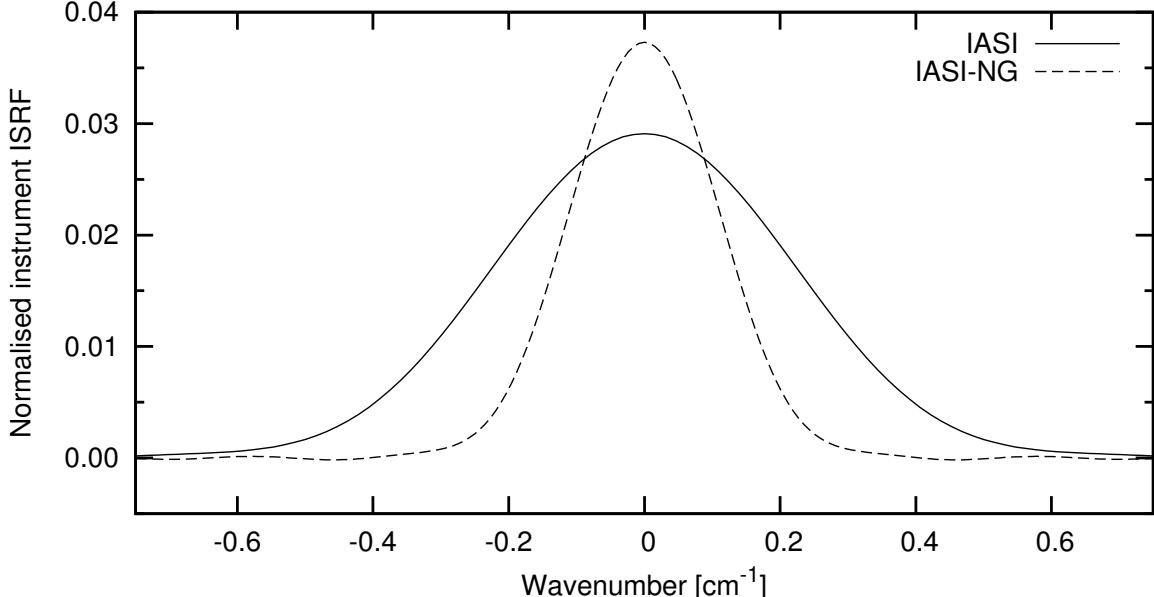

**Figure 2.** IASI and IASI-NG Instrument spectral response functions (ISRF). IASI-NG spectral resolution is doubled compared to IASI thanks to the optical path increment providing thinner instrument ISRFs.

An OpenMP parallel version of the 4A (Automatised Atmospheric Absorption Atlas)-OP 2012-1-1 model (Scott and Chedin, 1981; Tournier et al., 1995; Chaumat et al., 2012, http://ara.abct.lmd.polytechnique.fr/) and the RTTOV (Radiative Transfer for TOVS, TIROS (Television InfraRed Observational Satellite) Operational Vertical Sounder) model (Matricardi et al., 2004; Hocking et al., 2015) were chosen as RT models. Once the instrument data have been simulated, a random gaussian noise

5   using CNES specifications (Fig. 1) was added to IASI and IASI-NG data. Two different noises were used for the IASI-NG simulations according to the two prism materials currently under consideration.

## 3.1   Input data

As mentioned above, the most accurate description of the atmosphere is required to feed the RT models. In order to achieve this goal, the vertical profiles from a selection of atmospheric constituents in the different IR absorption bands measured by both

10   instruments were extracted from global analyses provided by the Monitoring Atmospheric Composition and Climate (MACC) project of the Copernicus programme[1]. The extracted vertical profiles were the profiles of temperature, specific humidity, $O_3$, CO and $SO_2$. They were provided in 60 fixed pressure levels from 1013.25 to 0.2 hPa. For consistency sake, the surface elevation, surface temperature and pressure were also obtained from this source.

---

[1]http://www.gmes-atmosphere.eu



The $CO_2$ and $CH_4$ three dimensional global fields used for observation to profile interpolation rely on the global tracer transport model LMDZ (Hourdin et al., 2006), driven by the wind analyses from the European Centre for Medium-Range Weather Forecasts, and using optimized surface fluxes following the configuration used by Chevallier et al. (2011). For $CO_2$ fields, the model has a horizontal resolution of 3.75 degrees in longitude and 1.9 degree in latitude with 39 vertical layers, from close to the surface up to 8.6 hPa. For $CH_4$ fields, the horizontal resolution is 3.75 degrees in longitude and 2.5 degrees in latitude with 19 levels, from surface up to 295 hPa.

A bilinear interpolation weighted by the inverse of the square distance was used to match the corresponding profiles and the surface values associated to each one of the IASI observations. The position and time of these IASI observations were extracted from the Météo-France operational archive. In addition to the latitude, longitude, date and time values of the IASI observations, the following parameters were also extracted from this archive: IASI azimuth and zenith angles, solar azimuth and zenith angles, cloud cover from the Advanced Very High Resolution Radiometer (AVHRR) on board Metop, land-sea mask values from the Météo-France global model ARPEGE and IASI measurements for a the subset of 314 channels selected by Collard (2007) and monitored operationally at Météo-France in 2013.

## 3.2 Radiative transfer models

4A is an optimized line-by-line (LBL) radiative transfer model (Scott and Chedin, 1981; Tournier et al., 1995; Chaumat et al., 2012) used as reference RT model for the CNES/EUMETSAT IASI Level 1 Cal/Val activities and operational processing. Using the OpenMP technology, a parallel-spectrum processing capability was added to the model in order to reduce the required computing time per spectrum. Rather than using the onerous full LBL models like LBLRTM (Clough et al., 2005) or STRANSAC (Scott, 1974), less expensive but not as accurate optimised LBL models have been developed. For typical instruments like IASI and IASI-NG, precision is better than the instrumental noise. 4A allows a fast computation of the transmittance of a discrete atmosphere along the vertical at a very high spectral resolution as well as the jacobians (Cheruy et al., 1995) (with respect to temperature, mixing ratios and surface temperature and emissivity) for a user-defined observation level. The model relies on comprehensive atlases of monochromatic optical thickness for up to 50 atmospheric molecular species and 43 pressure levels. The atlases were created using the line-by-line and layer-by-layer model STRANSAC in its latest version at the release date with spectroscopy information from the GEISA (Gestion et Etudes des Informations Spectroscopiques Atmosphériques, Management and Study of Spectroscopic Information) 2011 spectral line data catalogue (Jacquinet-Husson et al., 1999, 2003, 2011; Jacquinet-Husson, 2008). The 4A model also includes up-to-date continua of $N_2$, $O_2$ and $H_2O$. 4A uses surface emissivity values supplied by Snyder et al. (1998).

RTTOV (Matricardi et al., 2004; Hocking et al., 2015) is a fast RT model for passive visible, infrared (IR) and microwave (MW) satellite-borne sensors and is used in various applications including data assimilation, atmospheric retrievals and the generation of simulated satellite imagery. Fast RT models can reproduce LBL radiances with a good accuracy and computational efficiency that fulfils NWP requirements of near-real time monitoring and satellite radiance assimilation. This kind of models use computationally efficient parametrisations that allow them to simulate radiances at a fraction of the cost required by a LBL model.





RTTOV version 11.3 was used in this study. The latter takes as input vertical profiles of pressure, temperature, water vapour and optionally other trace gases, and scattering particle parameters along with associated surface parameters. The outputs are top-of-atmosphere (TOA) radiances, brightness temperatures (BTs) and the jacobians for the selected instrument channels. To carry out the fast computation of optical depths, RTTOV uses linear regressions to compute optical depths on a fixed set of pressure levels. The coefficients for these linear regressions are pre-computed and stored in coefficient files which are specific
to each instrument.

Most RTTOV coefficient files are based on a fixed set of 54 levels, from 1050.0 to 0.005 hPa, but coefficients have also been generated on a set of 101 levels, from 1100.0 to 0.005 hPa, for some instruments, more explicitly the IR hyperspectral sounders AIRS, IASI, CRIS, IASI-NG and IRS. RTTOV can accept input profiles on an arbitrary set of pressure levels. A vertical internal interpolator maps the input profile onto the coefficient levels and, again, the computed optical depths back
onto the input levels (Hocking, 2014). In the latest version of the software, RTTOV 11.3, additional options have been added to improve consistency of the input vertical profile units (Hocking et al., 2015).

### 3.3   Configuration of the simulations

The information contained in the IASI observation database has been used to prepare the most realistic possible atmospheric state according to the input of each RT model. Regarding the atmospheric chemistry, a total of 16 vertical atmospheric con-
stituent profiles were ingested by 4A for the simulations: $H_2O$, $CO_2$, $O_3$, $N_2O$, $CO$, $CH_4$, $SO_2$, $HNO_3$, $OCS$, $CH_3D$, $N_2$ and the CFCs 11, 12 and 14. For $H_2O$, $CO_2$, $O_3$, $CO$, $CH_4$, and $SO_2$ species, the vertical profiles were taken from the above database. For the other species, a standard profile provided by 4A was used. All the profiles were linearly interpolated onto the 43 4A levels (43L). The reduction in the number of levels was carried out as no significant gain in accuracy was observed when using 60 levels whereas the computing time is almost doubled in a single computer (evaluation against IASI observations not
shown). RTTOV simulations were run over 60 levels with the latest coefficient files and only 5 vertical profiles of atmospheric constituents were provided to the model: $H_2O$, $CO_2$, $O_3$, $CO$, and $CH_4$. The vertical profiles of only 6 atmospheric constituents can be supplied to the version 11 of RTTOV against 43 for 4A. The latest atmospheric constituent absent from the previous list is $N_2O$.

4A simulations have used surface emissivity values from internal tables over sea and sea ice. RTTOV uses ISEM model
(Sherlock and Saunders, 1999) for sea surface emissivities. Land surface emissivities were taken for 4A from the University of Wisconsin (UW) IR atlas of emissivities (Seemann et al., 2008) for the year 2013. RTTOV simulations also used the UW atlas of emissivities but for the year 2007. Since RTTOV uses precomputed atlas values to speed up the calculations, it was not possible to use the 2013 atlases. As negligible differences are expected in the UW atlas emissivity values between the years 2007 and 2013, the RTTOV precalculated emissivity files have been used. Table 2 summarizes the main differences between both RT runs.

The IASI viewing geometry was used for the simulations of both instruments despite the different scanning geometry of IASI-NG because scan geometry was not yet clearly defined at the time the dataset was built.





|  | 4A | RTTOV |
|---|---|---|
| Type | pseudo-LBL | Fast RT model |
| Levels | 43 | 60 |
| Gases from MACC | $H_2O$, $CO_2$, $O_3$, CO, $CH_4$, $SO_2$ | $H_2O$, $CO_2$, $O_3$, CO, $CH_4$ |
| Gases from RT model | $N_2O$, $HNO_3$, OCS, $CH_3D$, $N_2$ and CFCs 11, 12 and 14 | $SO_2$ |
| Sea surface emissivity | Snyder et al. (1998) | ISEM model |
| Land surface emissivity | UW Atlas 2013 | UW Atlas 2007 |

**Table 2.** Summary of the main differences between 4A and RTTOV radiative transfer (RT) model run configurations

5  Although the AVHRR cloud cover value corresponding to each IASI pixel and cloud vertical profiles have been included in the database, we have chosen not to consider cloudy conditions because of computational cost of these kind of simulations and also due to the large uncertainties in cloud radiative properties.

Once the simulations were carried out, a random gaussian noise were added to the simulations using the standard noise NedT at 280K provided by the CNES (fig. 1). Two different noises were considered for IASI-NG depending on the prism material 10  (KBr or ZnSe), which produced two IASI-NG simulation datasets.

## 4  Database results and validation

### 4.1  Information contained in the database

A total of 5,242,047 IASI observations were compiled for the four selected dates. From this total, 3,463,432 observations correspond to sea pixels, 185,916 were from coastal regions and 1,591,529 from over land. An overview of the distribution of 15  the observation according to different latitude bands and AVHRR cloud mask values is given in table 3. Five belts have been considered: the Northern Pole, the Mid-latitudes of the Northern Hemisphere, the Tropics, the Southern mid-latitudes and the Southern Pole. For the AVHRR cloud mask, there is no value for some observations. This absence of value is related to their production centre. These IASI observations, produced by *Centre de Météorologie Satellitaire (CMS)* of Météo-France and not by EUMETSAT from a direct broadcast have a much better timeliness compatible with specific NWP needs. However, at the time, they had not any cloud mask even if they are now provided with this information. They covered a wide area around





| Latitude band | No flag | Clear | P. cloudy | Cloudy |
|---|---|---|---|---|
| North Pole 70°N-90°N | 308 | 24,739 | 202,626 | 267,869 |
| NH Mid-lat 20°N-70°N | 82,402 | 311,466 | 601,094 | 598,305 |
| Tropics 20°S-20°S | 0 | 271,653 | 516,959 | 372,931 |
| SH Mid-lat 20°S-70°S | 0 | 159,580 | 540,066 | 799,668 |
| South Pole 70°S-90°S | 0 | 22,229 | 182,391 | 287,761 |
| Total | 82,710 | 789,667 | 2,043,136 | 2,326,534 |
| (%) | 1.6% | 15% | 39.0% | 44.4% |

**Table 3.** Geographical distribution of 5,242,047 IASI simulations (performed over four days in 2013) according to latitude bands with the corresponding AVHRR cloud mask value ("No flag" states that the cloud mask value is missing).

Brittany, where the CMS is located. The Northern Hemisphere presents more clear observations compared to the South. In

total, clear cases represent 15.0% of the total number of simulations. The no-flag, partly cloudy and cloudy simulations are respectively 1.6%, 39.0% and 44.4% of the 5,242,047 total simulations.

Figure 3 presents the information available in the database associated to an IASI observation taken out in the South Pacific (179.0 W, 24.55 S) on the February $3^{rd}$, 2013 at 21:13:46 UT. The simulated brightness temperatures presented in this Figure come from the 4A RT model dataset. Figure 3a compares the results from full IASI (black) and IASI-NG (grey) spectrum

simulations. IASI-NG presents a higher variability of the spectrum compared to IASI because of its higher spectral resolution. As a result, it is expected that new atmospheric constituents can be detected by IASI-NG and that those already detected, such as water vapour isotopologues, will be better characterised (Wiegele et al., 2014).

The differences between the IASI 4A simulated spectrum and the corresponding observation for the 314 available channels in the Météo-France archive range from -3 to 3 K except the region between 2,280 and 2,400 cm$^{-1}$ where they increase

to around 7 K (Figure 3b). Those bands associated to the input vertical profiles present larger differences than the regions corresponding to surface properties. The region between 2,280 and 2,400 cm$^{-1}$ is strongly affected by the so-called *Ghost effect* (Lezeaux, 2007; Bormann et al., 2010). This effect was caused by the IASI compensation device but vanished when the device was turned off on October 7th, 2015 (Maraldi et al., 2015b). Finally the third horizontal panel presents the sea surface emissivity from the 4A model. In this case, the surface emissivity varies between 0.971 and 0.991.

The first row of Figure 3d presents the values of temperature and cloud properties extracted from the MACC project. The vertical profiles of the different atmospheric constituents used for the simulation are shown in Figure3e. The additional information concerning this IASI observation can be found in Table 4: latitude, longitude, terrain elevation, date, time, surface pressure and temperature, ARPEGE land-sea-mask, AVHRR cloud cover, instrument zenith and azimuthal angles and solar zenith and azimuthal angles.





**Figure 3.** Example of the information for a single simulation corresponding to $3^{rd}$ February 2013 at 21:13:46 : the IASI and IASI-NG simulations (a), the differences between the 4A IASI simulations and the IASI observations (b) and the surface emissivity spectrum (c). The bottom panel (d) displays the vertical profiles of temperture, specific humidity, cloud cover, hydrometeor contents (Ice water content (IWC), Liquid water content (LWC), and atmospheric trace gases.





| Parameter | Value |
|---|---|
| Latitude | 24.55 S |
| Longitude | 179.04 W |
| Elevation | -0.05 m |
| Date | 03/02/2013 |
| UT Time | 21:13:46 |
| Surface Pressure | 1007.79 hPa |
| Surface temperature | 299.28 K |
| Land-Sea-mask value | 0.0 |
| AVHRR Cloud cover | 0 |
| IASI Zenith Angle | 8.73 |
| IASI Azimuth Angle | 287.49 |
| Solar Zenith Angle | 41.94 |
| Solar Azimuth Angle | 87.28 |

**Table 4.** Additional parameters associated to the observations shown in Figure 3

## 4.2 Evaluation of simulations results

A first evaluation of the simulations was made using the IASI MetOp-A full spectrum brightness temperatures from the orbit beginning 6 August 2013 at 20:35:59 considering only clear-sky observations over sea including sea-ice during night-time. Hence, 2877 out of the 91.800 IASI measurements carried out in each IASI orbit were retained.

Table 5 presents the mean and standard deviations of the differences between 4A and RTTOV simulations against IASI brightness temperatures for the selected satellite orbit. No differences have been found in terms of standard deviations for the two models. On the other side, the mean values (biases) values present differences that are displayed in Figure 4. In the first band, the RTTOV bias is lower than the 4A one excluding the $CO_2$ band, between 665 and 675 cm$^{-1}$ and from 700 to 745 cm$^{-1}$. It should be noted that RTTOV uses a set of predefined $CO_2$ to generate the coefficients file which dates from 2013 which varies between 300 and 500 ppmv. Since that date, the global $CO_2$ concentration has grown up by 2.5% approximately, possibly outdating this RTTOV profile training set. 4A computes optical depths directly using the input vertical profiles allowing a more accurate values for them than using a set of predefined lookup tables. The reason of RTTOV for presenting a lower bias than 4A in the rest of the band arises from not having considered the same sea surface model. As mentioned before, 4A uses constant emissivity values from Snyder et al. (1998), while in RTTOV the sea surface emissivity is computed by the ISEM model, which takes into account the skin temperature and the instrument viewing angle (Hong et al., 2010).

Whatever the spectral bands, differences could be due to the spectroscopic parameters used. The spectroscopy used by 4A came from the 2009 version of GEISA database and has not been intentionally updated in the moment of carrying out the simulations, in comparison with RTTOV where HITRAN 2012 has been used. In the second IASI band, RTTOV bias values



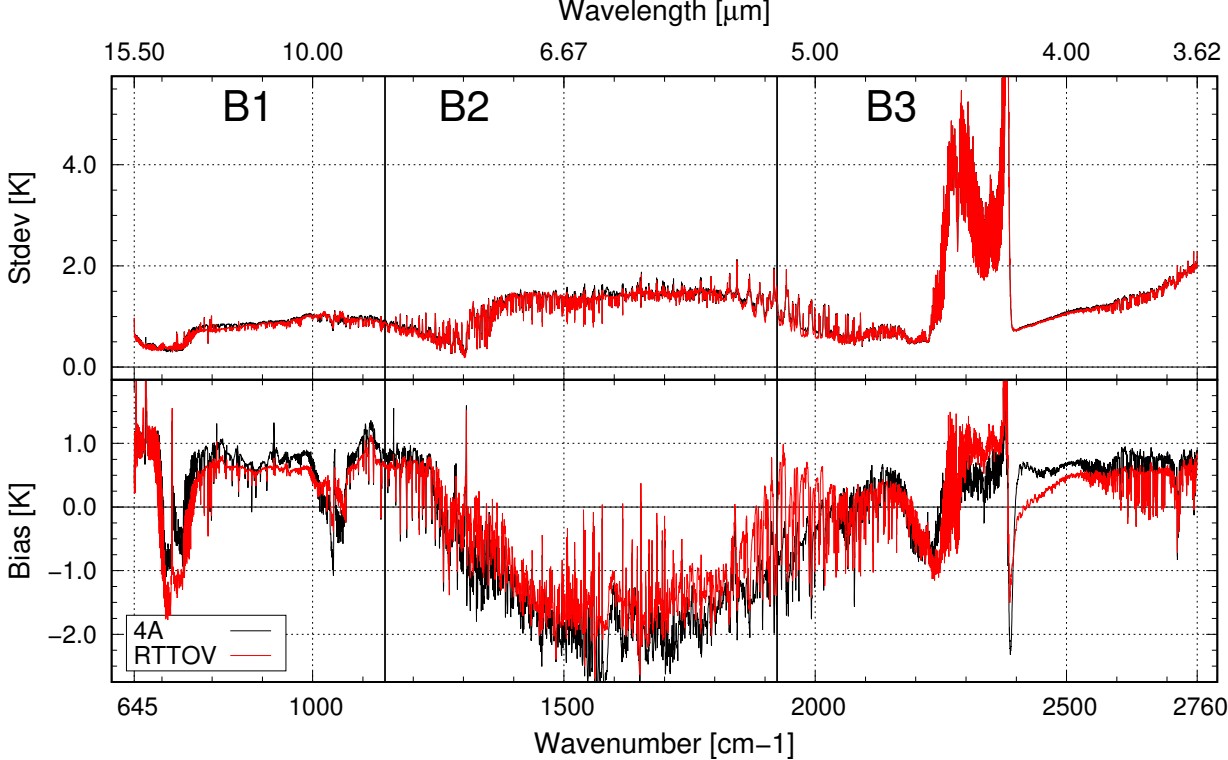

**Figure 4.** Statistics of IASI brightness temperature differences between model simulations (4A and RTTOV) and observations in terms of standard deviation (top panel) and bias (bottom panel). The observations have been sampled during night-time over clear sky ocean surfaces for a single orbit (6th of August 2013). The sample size is 2877. B1, B2 and B3 correspond to the three IASI bands.

| Band | 4A | | RTTOV | |
|---|---|---|---|---|
| | Bias | SD | Bias | SD |
| 1 | 0.56 | 0.21 | 0.40 | 0.21 |
| 2 | -1.22 | 0.34 | -0.89 | 0.34 |
| 3 | 0.22 | 1.13 | 0.23 | 1.13 |

**Table 5.** Bias and standard deviation (SD) of IASI brightness temperature differences between 4A and RTTOV simulations and observations for the sample shown in Figure 4 (expressed in Kelvin (K)).

are equal or lower to those of 4A. These differences arise most of the time from the different representation of the water vapour continuum at the stage of the simulations.

10     In band 3, the RTTOV bias and standard deviations are closer to zero at the beginning of the band, but present larger values than 4A in the emission $CO_2$ spectral window from around 2,220 to 2,380 $cm^{-1}$. In the solar region, above 2,400 $cm^{-1}$, the 4A bias remains constant around 0.7-0.8 K whereas RTTOV present lower bias values. The origin of these differences could




come from: i) the different sea surface emissivity values used by the two models, ii) the $N_2$ continuum and ii)) line mixing effects around 4.3 $\mu$m.

The up to 10 K peak observed in the standard deviation curves of both models at 2,390 cm$^{-1}$ from Figure 4 is mainly caused by the so-called Ghost effect (Bormann et al., 2010). It is generally acknowledged that this effect emanates from a perturbation in the response of the instrument which is mainly caused by micro-vibrations of the interferometer separator blade, in turn induced by the instrument compensation device (Maraldi et al., 2015a). As it has been found that IASI reconstructed radiances from a principal component compression do not exhibit this artefact (Hultberg, 2010). To confirm this statement, we have undertaken an additional comparison between the simulated spectra and reconstructed radiances has been carried out (Figure not shown). Although a difference of 10K in the standard deviation was found for IASI channel 6942, the value decreases to 2.0K with reconstructed radiances. This confirms that the peak of 10K in the standard deviation comes from the verification observation and not from the RT model.

In order to evaluate both model simulations over a longer period (the four selected days in 2013), simulations were compared against a subset of 314-channels of IASI observations stored at Météo-France archive. Only 789,573 clear sky observations over both land and sea from the 5,242,047 total number of observations, i.e., around a 15% of the recovered IASI observations were cloud-free observations according to the AVHRR cloud cover. The average and standard deviation of these differences for both 4A and RTTOV simulated datasets are shown in Figure 5.

The standard deviations of 4A and RTTOV simulations against observation differences are similar all over the measured spectrum. There are only three slight differences: for surface channels in the range between 750 and 950 cm$^{-1}$, RTTOV presents a slightly lower values compared to 4A: around 0.11 K in average, a reduction of about 3.2% with respect to 4A. On the contrary, in the region between 1020 and 1225 cm$^{-1}$, the standard deviations of RTTOV differences are in average 3.5% higher than with 4A (around 0.1 K). The third area, where differences between RTTOV and 4A are noticed, is in the solar part of the spectrum, above 2400 cm$^{-1}$. These larger standard deviations for RTTOV come from the fact that, contrary to 4A, the solar contribution is ignored.

Regarding the differences in biases, RTTOV exhibits values closer to zero than 4A, excluding the $CO_2$ band between 700 and 800 cm cm$^{-1}$, the surface sensitive region between 1100 and 1200 cm$^{-1}$, and the three channels above 2400 cm$^{-1}$, probably due to the differences in the UW atlas surface emissivity values used. As it was discussed previously, emissivity values corresponding to the year 2007 were used for RTTOV simulations instead those of the year 2013.

### 4.3 Differences between IASI and IASI-NG simulated spectra

A comparison between IASI and IASI-NG radiance simulations is shown in Figure 6, from 730 to 740 cm$^{-1}$, belonging to the 15 $\mu$m $CO_2$ absorption band. A single IASI observation from 6 August 2013 and acquired at 20:38:00 has been used in order to calculate the Earth IR spectrum with 4A at a resolution of 0.001 cm$^{-1}$. The result is shown Figure 6b. Figure 6a presents the $CO_2$ absorption lines as extracted from the GEISA database (Jacquinet-Husson et al., 2011). The position of these absorption lines corresponds with the sharp peaks observed in the 4A spectrum calculation of Figure 6b.

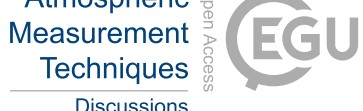



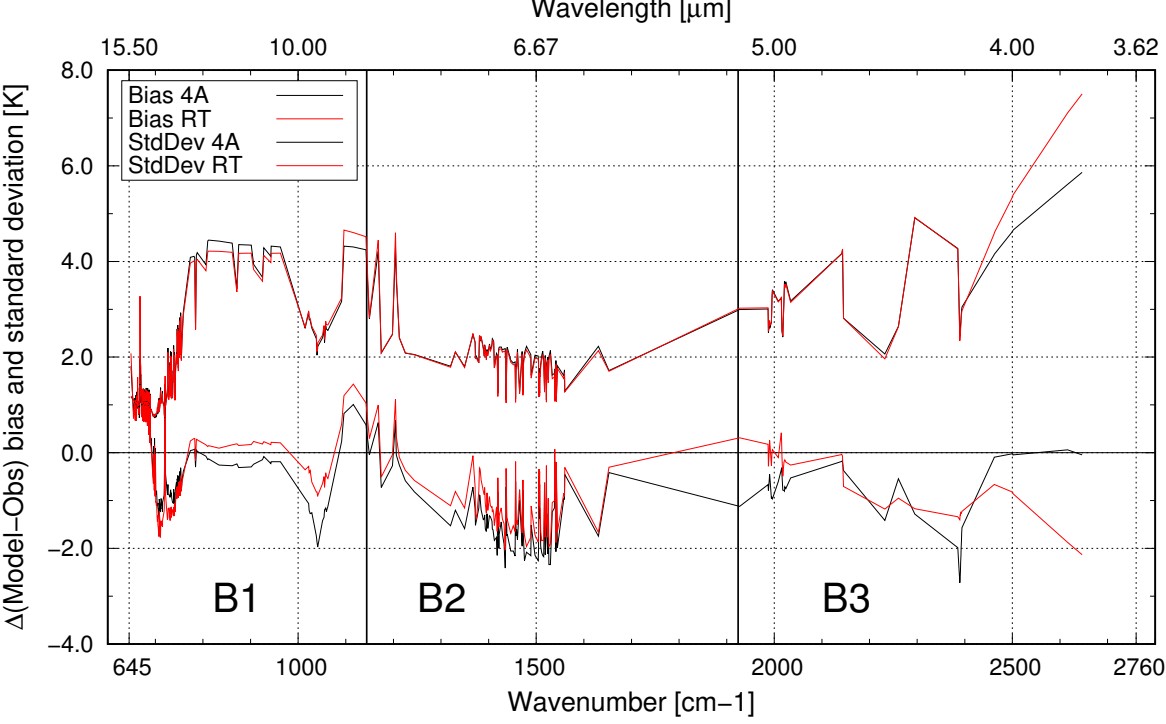

**Figure 5.** Bias (straight lines) and standard deviation (dashed lines) of differences between model simulations and observations in 314 channels for the 789,573 IASI observations taken under clear-sky conditions according to the AVHRR cloud flag during daytime and nighttime and above all surface types. B1, B2 and B3 correspond to the three IASI bands.

The simulation of the IASI and IASI-NG spectrum by RTTOV and 4A models is shown in Fig 6c and d respectively. The values of the corresponding IASI measurements are also drawn, with dots, in Figure 6c. The simulations of IASI and IASI-NG spectrum from both models are very close, almost undistinguishable in this spectral range. Additionally, the differences between the measurement and the IASI simulations are very small showing the quality of the two models in this spectral window. IASI-NG signal presents a higher variability giving values going from 0.05 to 0.09 W/m$^2$sr/cm$^{-1}$ for the 0.058 to 0.085 W/m$^2$sr/cm$^{-1}$ of the IASI range. This higher variability will increase the low-concentration limit of trace gas detection of IASI-NG compared to IASI.

## 5 Retrieval of temperature and humidity vertical profiles using the simulated observation database

In this section, retrieval experiments are performed using the 1D-Var code (version 1.0) provided by the EUMETSAT NWP Satellite Application Facility (Weston, 2014). This 1D-Var system was interfaced with the RTTOV model version 11. A subset of 1,681 4A simulations was extracted from the global database presented in previous sections. This subset retained only one simulation over sea, clear-sky observations according to the AVHRR cloud mask flag for each date evenly distributed over





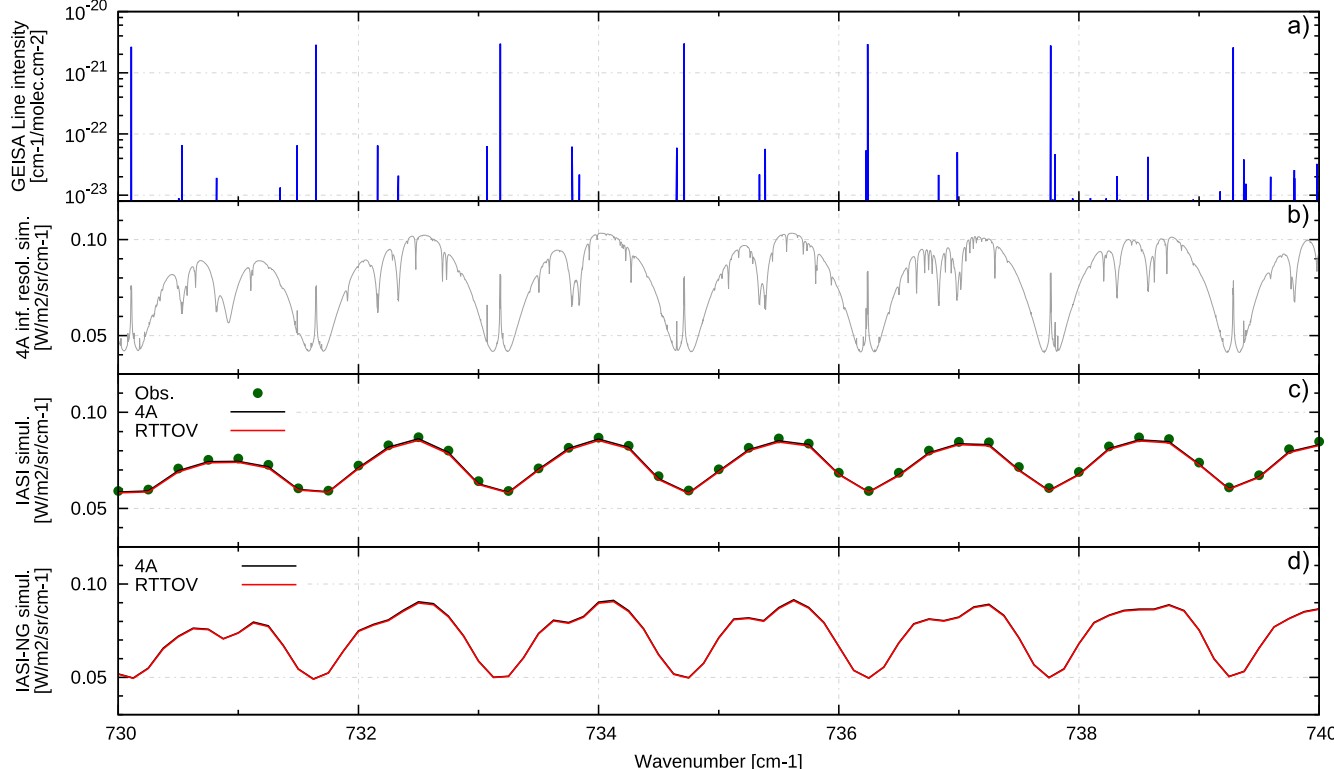

**Figure 6.** Comparison of IASI/IASI-NG radiance simulations in the 730-740 cm$^{-1}$ $CO_2$ absorption window against the Earth IR spectrum computed by 4A using a spectral resolution of 0.001 cm$^{-1}$ for one atmospheric description included in the simulation dataset. Upper panel (a) presents the $CO_2$ absorption lines as obtained from GEISA database, the second panel (b) shows the 4A spectrum calculation. Third and four panels present the simulations of IASI (c) and IASI-NG spectra (d). IASI observations are also included in the IASI simulations panel (green dots).

latitudes and longitudes. Observations above sea ice surface, mainly in the South Pole, were excluded from the dataset. These 1681 simulations were classified by latitude bands (Figure 7), resulting in 522 observations for tropical latitudes (20S to 20N), 1068 for mid-latitudes (66.5S to -20S and 20N to 66.5N) and 91 for polar latitudes (90S to 66.5S and 66.5N to 90N).

## 5.1  1D-Var framework

The background error covariance B matrix provided by the package assumes an a priori information of a very high quality, i.e.,
5  an error on temperature retrievals of 0.5 K when IASI is supposed to be able to improve temperature retrieval errors below 1K. To be consistent with the IASI specifications the B matrix errors have been multiplied by 2. Background profiles were obtained from the true atmospheric state perturbed by a noise corresponding to the B matrix. For the observation error-covariance R matrix, a diagonal matrix formed by the Météo-France operational-specified IASI errors was used. As the noise of IASI-NG is





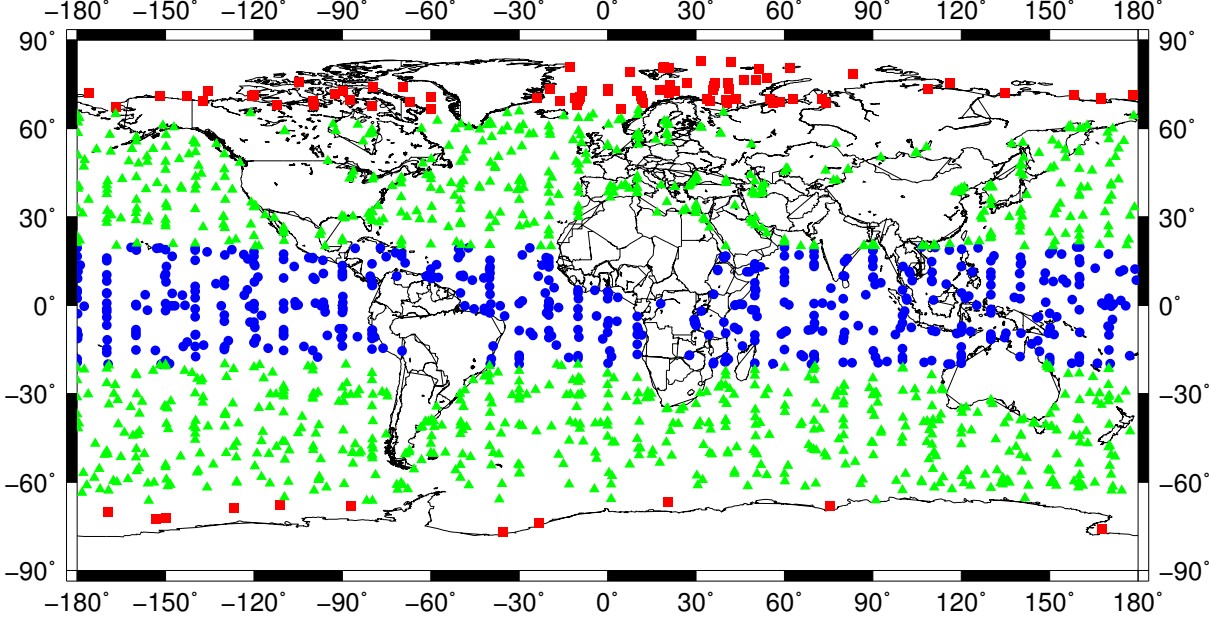

**Figure 7.** Localizations of the 1681 simulations considered for running 1D-Var inversions by latitude bands: 91 for polar latitude in red squares, 1,068 for mid-latitudes in green triangles and 522 for tropical latitudes in blue dots.

10  half of IASI, the R matrix values were divided by 4. This operational R matrix overestimates mostly the errors for water vapour channels as the inter-channel correlations in their errors were not taken into account.

We have chosen the current operational channel selection of the ARPEGE model, consisting of 123 IASI channels out of the total 8461. For IASI-NG, the channel selection included the 123 channels corresponding to the same wavenumbers than those of the IASI channel selection (see Table 6 for further details). This channel selection covers the whole atmosphere for temperature profile and only the troposphere for the humidity (Figure 8) even though the jacobians peaks slightly differ between both instruments. These jacobians have been obtained with RTTOV, and correspond to jacobians of both sounders

5  averaged over the 1,681 simulations. Humidity jacobians depend on the water vapour profile, presenting higher values for higher humidity concentration (not shown). In the Tropics, the humidity jacobian peaks at higher altitudes in the atmosphere compared to polar regions because of this dependency with the water vapour concentration.

A bias correction was applied to each latitude-band experiment. The values of this correction were obtained after a first run of the 1D-Var retrieval. The average difference between the first guess and the simulated brightness temperatures were removed

10  from the observations to eliminate this bias for a second 1D-Var run.

To analyse the retrievals from the 1D-Var runs, the standard deviation of the observation minus background and observation minus retrieval differences were computed for the brightness temperatures. For temperature and humidity vertical profiles, the standard deviation computed were those of the truth-background and truth-retrieval differences. The analysis of the results



| Channel type | N | Channels |
|---|---|---|
| Stratospheric T channels | 37 | 49, 51, 55, 57, 59, 61, 63, 66, 79, 81, 83, 85, 87, 104, 109, 111, 113, 116, 122, 125, 128, 131, 133, 135, 138, 141, 144, 146, 148, 151, 154, 157, 159, 161, 163, 167, 170 |
| UTLS channels | 39 | 173, 176, 179, 180, 185, 187, 193, 199, 205, 207, 210, 212, 214, 217, 219, 222, 224, 226, 230, 232, 242, 254, 260, 267, 269, 275, 280, 282, 294, 296, 299, 303, 306, 323, 329, 354, 360, 366, 386 |
| Mid-tropospheric T channels | 8 | 265, 345, 347, 350, 356, 373, 375, 383 |
| Low-troposphere T channels | 14 | 327, 398, 401, 404, 407, 410, 414, 426, 428, 432, 434, 439, 445, 457 |
| Surface channels | 4 | 515, 1191, 1194, 1271 |
| Mid and high trop. Q channels | 14 | 2701, 2910, 2951, 2958, 2991, 2993, 3002, 3008, 3014, 3027, 3049, 3058, 3105, 3577 |
| Low-trop. Q channels | 7 | 5368, 5383, 5397, 5401, 5403, 5405, 5483 |

**Table 6.** List of the 123 IASI channels used for the profile retrievals.

consisted in comparing the ratio of standard deviations of the differences versus the retrievals compared to the differences versus the background:

$$\sigma_{reduction} = \frac{\sigma_{obs/truth-ret}}{\sigma_{obs/truth-bck}} - 1 \tag{1}$$

Negative values of $\sigma_{reduction}$ means a reduction on the retrievals standard deviation, hence, a positive impact, whereas positive values of this index correspond to a negative impact or an increase in the retrievals standard deviation.

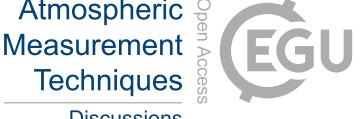



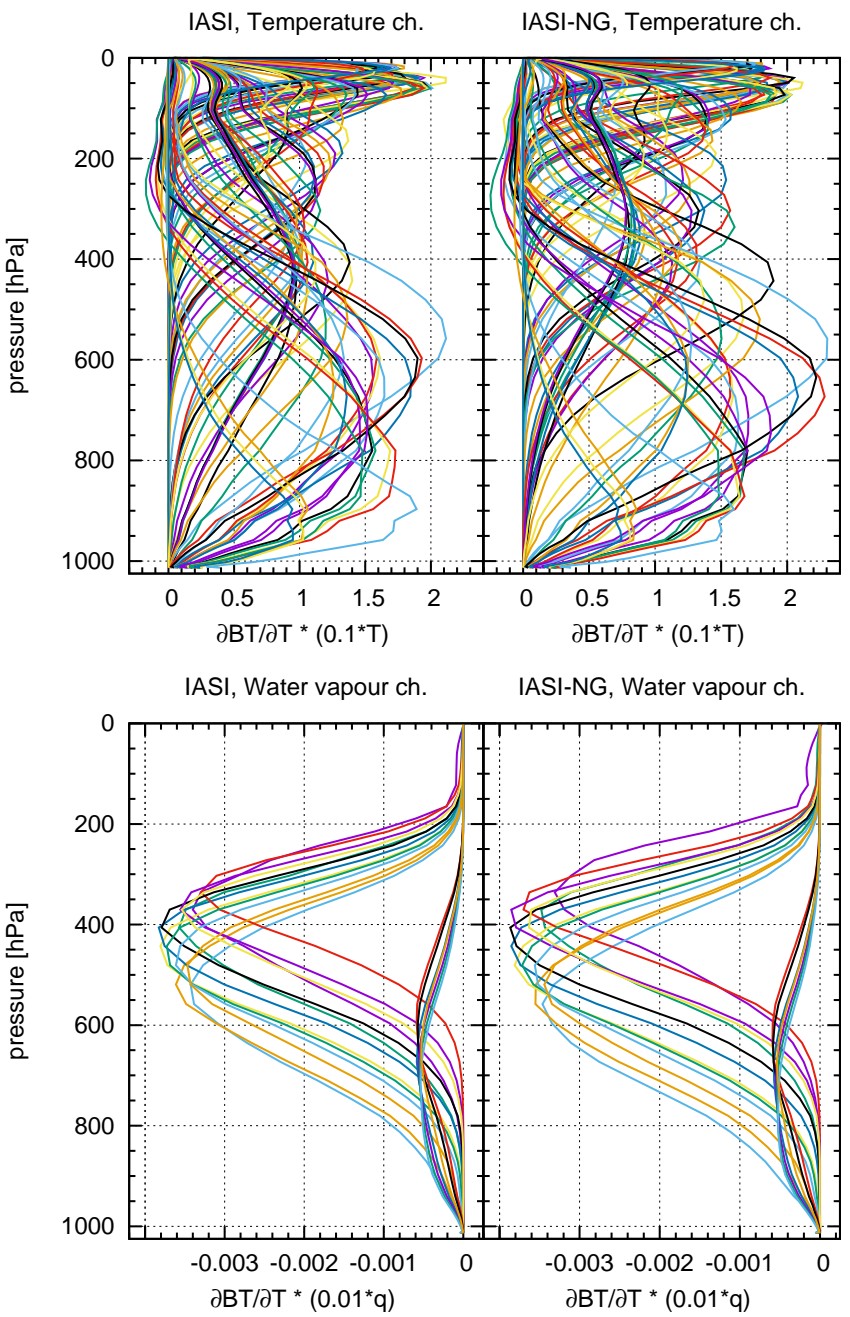

**Figure 8.** The 123-channels temperature (top panels) and humidity (bottom panels) averaged jacobians from the considered 1681 simulations and computed for IASI (left hand side panels) and IASI-NG (right hand-side panels).





## 5.2 Results

Figure 9 presents the values of the $\sigma_{reduction}$ index for the observations for the three latitude bands. High stratospheric channels present an improvement in the error reduction of a 0.3 value, meanwhile the differences for tropospheric-peaking channels is better only by a 0.1 in average. This difference is caused by the strong overestimations of the instrument noise for humidity channels because of the strong intercorrelation for these channels.

Similar plots to Figure 9 are presented in Figures 10 and 11 but for the error reduction in temperature and humidity profiles respectively. We found a difference of up to a 10% in the reductions of temperature profile error for the tropical band around 400 hPa. Mid-latitude and polar latitude bands present lower error reduction differences, around 5% in the troposphere, with a flatter shape in average. The more striking shape of the polar latitudes error reduction profile could have its origin in the lower number of used profiles (91 in polar latitudes for 1068 in mid-latitudes). The error reductions in the stratosphere are lower than those observed in the troposphere, but IASI-NG has still higher error reductions than IASI. Apart from the first kilometres of the atmosphere, IASI-NG displays an error reduction twice higher than that given by IASI up to an atmospheric pressure around 750 hPa. The worst performance of both instruments in the first atmospheric layers is related to the lack of sensitivity of IASI channels at these levels to the first atmospheric layers in the selection used in Météo-France operational system. A new channel selection for IASI-NG shall be carried out including channels able to improve this lack of sensitivity, and taking into account the IASI-NG bands 3 and 4 thanks to the IASI-NG noise reduction compared to IASI.

The humidity error reduction profiles provided by both instruments present higher values than those found for the temperature error reduction profiles, where, in the best case, a 24% error reduction was reached. The IASI-NG error reduction profile presents higher error reduction values, of around 5%, from around 850 hPa or thereby to the tropopause compared to IASI. As it was observed for temperature profiles, the error reductions in the low troposphere are smaller than those found for mid-troposphere or the upper atmospheric layers. In addition to the explanation given for temperature profiles, it must be noted that the noise added to humidity background profiles in the low troposphere is smaller than the one considered in the other layers of the atmosphere.

A similar experiment to the IASI/IASI-NG experiment has been carried out to compare the performances of the two IASI-NG prism materials. Although a better error reduction was found for brightness temperature stratospheric channels, no impact was noticed in temperature or humidity profile retrievals. It should be noted that the channel selection used for this experiment was equivalent to the IASI one, just taking the channels corresponding to the same wavenumbers. In order to investigate the differences between both materials, a new channel selection, adapted to IASI-NG characteristics, shall be performed.

## 6 Conclusions

A database of IASI and IASI-NG simulations has been devised. The two sets of simulations for IASI-NG correspond to the two materials under consideration for the IASI-NG prism The database is presented in two versions according to the radiative transfer model that has been used. The information available in the database has the following structure:




- **Coordinates:** Longitude, latitude, terrain elevation, date and time.

30    – **Observation parameters:** Surface pressure and temperature, IASI zenith and azimuth angles, solar zenith and azimuth angles, land-sea mask value, cloud cover value from AVHRR, 4A emissivity index, surface emissivity from UW atlas and RTTOV surface type.

- **Vertical profiles:** Temperature, humidity, carbon dioxide, ozone, carbon monoxide, methane and sulphur dioxide.

- **Cloud information:** Vertical profiles of cloud cover, ice water content, liquid water content, rain water content and snow
water content.

- **Simulations:** Radiances of IASI, IASI-NG A (KBr) and IASI-NG B (ZnSe).

- **Observations:** 314 IASI brightness temperatures.

Two kinds of validations have been presented. For one single orbit, all clear cases over sea and during night-time, the RTTOV and 4A simulations have been compared to the measured IASI 8,461 channels. Similar results were found for the two models.
On one hand by using the sea surface emissivity ISEM model, RTTOV has slightly better results than 4A. On the other hand, land-surface emissivities from 4A are better because of using UW emissivities for the 2013 instead of 2007 like RTTOV. RTTOV appears to better represent the water continuum providing a lower bias value in IASI band 2.

By simply considering the noise reduction of IASI-NG the improvement on brightness temperature error reduction mainly varies between 5 and 15 percent compared to IASI. A preliminary study of the benefit of IASI-NG compared to IASI has been
presented. With channels located in the same wave numbers, retrieval experiments showed an improvement of the temperature retrievals along all the atmosphere with a maximum in the troposphere. The improvement is lower for the humidity in the troposphere. A reduced sensitivity in the low troposphere is confirmed for IASI and, additionall work is required to check if IASI-NG will be able to better probe the atmosphere at these levels. For this purpose, a new channel selection needs to be defined, which will be undertaken in a following study.

## 7   Code availability

A high level Fortran 90 library has been developed to access the information contained in the simulation dataset files. The library and its documentation can be downloaded from the web page of the EUMETSAT/CNES IASI Sounding ScienceWorking Group at http://iasi.cnes.fr/en/IASI/isswg.htm.

## 8   Data availability

A total of 96 hourly files for the two simulation dataset are available at the web page of the EUMETSAT/CNES IASI Sounding
Science Working Group at http://iasi.cnes.fr/en/IASI/isswg.htm.





*Acknowledgements.* Jean Maziejewski and Jean-Francois Mahfouf are warmly thanked for their careful review of a previous version of the paper.




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





**Figure 9.** Error reduction ($\sigma_{reduction}$) in percentage for IASI observations in brightness temperature units for Polar regions (top), mid-latitudes (middle) and Tropics (bottom). Black and red curves present the error reductions for IASI and IASI-NG respectively, whereas the blue line means the differences between the both instruments. Coloured boxes correspond to the sensitivity of the channels (red=temperature, green=window, blue=water-vapour).





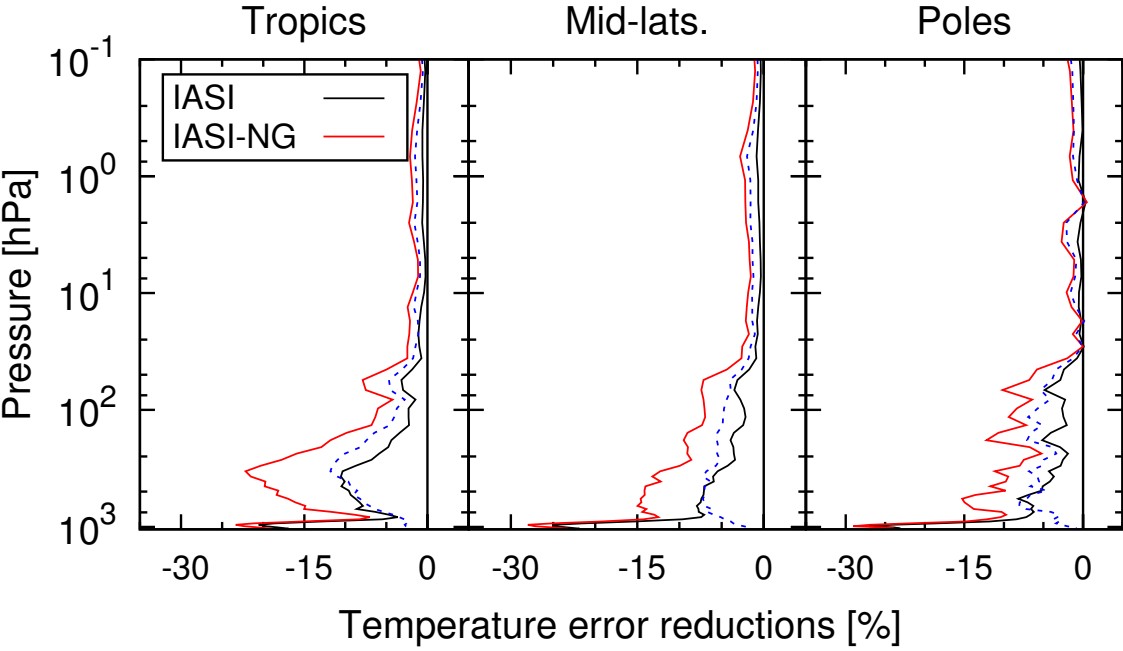

**Figure 10.** Error reduction ($\sigma_{reduction}$) in percentage with respect to the atmospheric pressure for temperature with the assimilation of the 123 channels for IASI (Black line) and IASI-NG (red line) with respect to regional areas (Tropics, mid-latitudes and polar regions). The blue dashed line corresponds to the error reduction difference between IASI-NG and IASI assimilation.





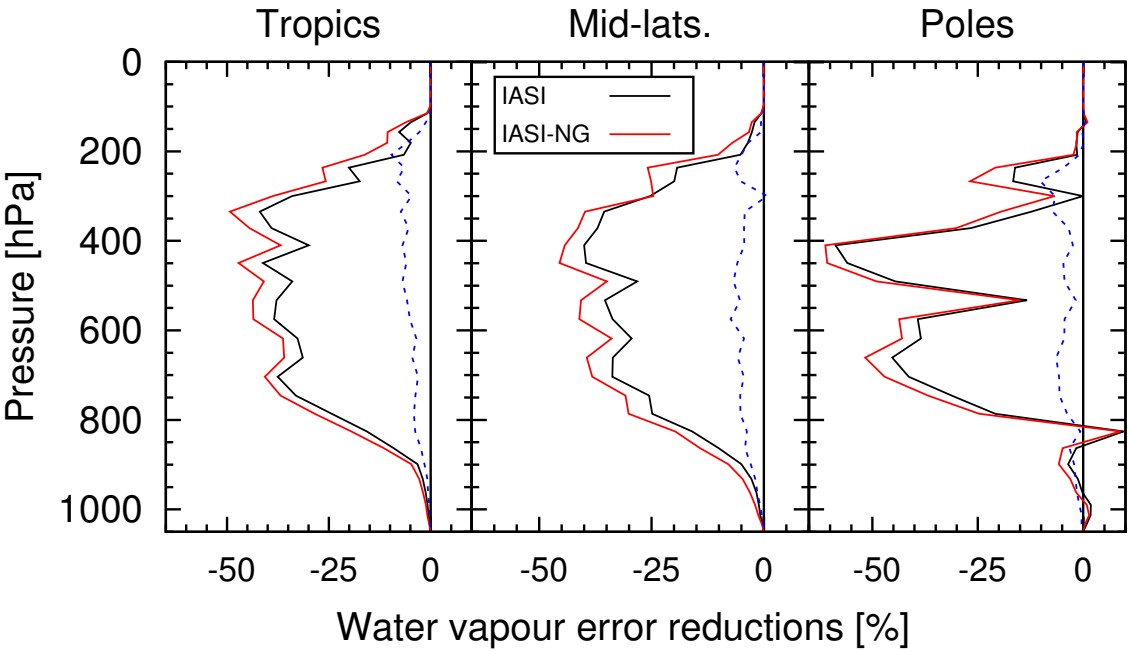

**Figure 11.** Error reduction ($\sigma_{reduction}$) in percentage for water vapour with the assimilation of the 123 channels for IASI (Black line) and IASI-NG (red line) with respect to regional areas (Tropics, mid-latitudes and polar regions). The blue dashed line corresponds to the difference of error reduction between IASI-NG and IASI assimilation.