# Peer review of "A simulated observation database to assess the impact of IASI-NG hyperspectral infrared sounder"

_Atmospheric Measurement Techniques, 2017_

## Referee Comment (RC1) · Anonymous Referee #2 · 31 Oct 2017

Review:

This paper provides a relatively straightforward analysis of the differences in performance between IASI and IASI-NG. This is of interest to the Earth Observation community. However, I think the paper as it stands is not suitable for publication in AMT. This is because I think the authors need to address the representativeness of their results, so that the scientific community can assess how, in general, IASI-NG is an improvement on IASI. I provide details in the specific comments below. The authors should also address the other specific comments.

Specific comments:

P. 1

L. 21: Please provide more details than Âńexperimental purposesÂż.

P. 2

L. 1: Check grammar, here and elsewhere: ". . .much of the information is. . .".

L. 6: Perhaps provide more details of the "conservative" approach.

L. 8: Usefulness of the channels for what?

L. 10: What is the operational mission?

L. 14-15: ". . .Medium-range Weather Forecasts".

L. 17: Impact on what parameter? What other data did the Met Office assimilate?

L. 19: What large-scale variables?

L. 28: What other instruments?

L. 33: I suggest you replace "race" with "effort".

P.3

L. 1: I would suggest you use a word different from "figures".

L. 3 (and elsewhere in this paragraph): OSSEs.

L. 8: Perhaps write: ". . .typically mimic. . .".

L. 11-13: Explain the advantages of using two different radiative transfer models.

L. 17: last -> latter.

L. 34: Larger noise than what?

P. 4

L. 6: If you use British spelling, it should be "programme".

L. 17: What is an adequate refractive index?

L. 17-18: I suggest you rephrase "The noise requirements... two."

P. 5

Table 1: There seems to be a missing entry (NedT for IASI).

Sect. 3: The database uses one day per season. Could you discuss if this is enough to be representative of seasonal conditions?

P. 6

L. 5: Provide more details of how you apply the noise. There are four NedT curves in Fig. 1.

L. 11: You use the MACC data for each of the four dates you mention previously in the paper?

P. 7

L. 29: Provide examples of this use of RTTOV.

P.8

L. 22-23: I do not understand this phrase.

P. 9

L.7: Perhaps the authors could provide a reference for the statement about uncertainties of cloud radiative properties.

L. 15: "...five latitude belts...".

P. 10

L. 7: What is the point of Fig. 3?

P. 12

L. 9: Introduce acronyms like ppmv. Grown up -> increased.

L. 10: Clumsy use of "outdating". Reword.

P.14

L. 5-6: Clumsy construction; please rephrase.

P. 19

Fig. 8: Indicate the meaning of the colours in the plots.

P. 20

L. 5: Avoid subjective words like "striking".

L. 17: This explanation is not very clear to me. Please clarify.

P. 21

L. 13+: As I see it, this paper shows that IASI-NG performs better than IASI. Is this to be expected? I presume that the value of the paper is that you quantify this improvement. How representative is this improvement? I would suggest that you discuss these points in the conclusions section.

---

## Short Comment (SC1) · 20 Nov 2017

Dear Authors,

please note that the following paper discusses the added-value of one possible configuration of IASI-NG in the characterisation of the lower troposphere in terms of the ozone concentration: Sellitto, P., Dufour, G., Eremenko, M., Cuesta, J., Dauphin, P., Forêt, G., Gaubert, B., Beekmann, M., Peuch, V.-H., and Flaud, J.-M.: Analysis of the potential of one possible instrumental configuration of the next generation of IASI instruments to monitor lower tropospheric ozone, Atmos. Meas. Tech., 6, 621-635, https://doi.org/10.5194/amt-6-621-2013, 2013. Even if your paper does not directly

address the topic of air quality, I think that citing this work would be useful when discussing your results (and IASI-NG expected added-value), see e.g. this sentence in your conclusions: "A reduced sensitivity in the low troposphere is confirmed for IASI and, additional work is required to check if IASI-NG will be able to better probe the atmosphere at these levels."

In addition, I also suggest to open your discussion to possible multi-spectral synergies, with reference to the following paper: "Costantino, L., Cuesta, J., Emili, E., Coman, A., Foret, G., Dufour, G., Eremenko, M., Chailleux, Y., Beekmann, M., and Flaud, J.-M.: Potential of multispectral synergism for observing ozone pollution by combining IASI-NG and UVNS measurements from the EPS-SG satellite, Atmos. Meas. Tech., 10, 1281-1298, https://doi.org/10.5194/amt-10-1281-2017, 2017."

I suggest adding these two references to put your very useful work in a slightly wider context.

My best regards,

Pasquale Sellitto

---

## Referee Comment (RC2) · Anonymous Referee #1 · 21 Nov 2017

Overview:

This paper documents the production of a set of databases designed to simulate IASI and IASI-NG radiances for future use in retrieval and data assimilation studies. It describes in detail how the origin of the "true" atmospheric states used in the calculations and the similarities and differences between the two radiative transfer models employed - RTTOV and 4A.

The paper achieves this goal well and (with some minor suggestions outlined below) can be accepted for publication based on that - although the amount of truly new science is limited.

[Figure]

The final section presents a somewhat simplistic evaluation of the relative retrieval skill from IASI and IASI-NG. I am assuming it is being presented as an example of the sort of thing that could be done with the database, but I do not think it necessarily adds much to the paper.

Detailed comments:

p.2 lines 13-14: I don't think you should say that "some channels were unsuited" as the real issue is not the channels themselves but that the information is redundant. Maybe say " a subset of channels is preferred"

p.4 line 18: "an adequate refractive index" -> "an appropriate refractive index"

p.6 lines 4-6: Is the noise added diagonal or is the fact that noise is correlated between channels because of apodisation allowed for? In fact, apodisation is only briefly mentioned - you should state explicitly the apodisation being used.

p.9, lines 18-20: I think the details on how you do not have some cloud flags because of some processing quirk are confusing and not really relevant.

p.12, lines 4-6 and Table 5: I do not think that Table 5 is particularly useful as it aggregates the information in Figure 4 while hiding much of the detail. It is also misleading to say there are no differences in the standard deviations when it is clear from Figure 4 that they do indeed exist. So I would remove this table.

Figure 6: Why did you choose to plot this in radiance units when everything else is in brightness temperature? It makes it very difficult to compare with the other plots.

Section 5: I think you should make it clear from the start that the retrieval discussion is illustrative. The scene selections (totally clear columns), channel selection (limited to IASI 314); background error (arbitrarily set to 2x the NWPSAF supplied one); simple bias correction and simple assumption that IASI-NG noise is exactly 2x IASI are probably not close to how that data will be used in practice.

p.15, lines 28-30: Please state explicitly that the 4A calcuations are going to be referred to as "truth" for the rest of the section.

p.17, line 9: Is it really true that the IASI-NG noise is half of IASI noise for all channels?

p.20, lines 9-13: While the channel selection could well be improved, you should be clear that there is a fundamental issue with very low level retrievals in the infrared due to the lack of contrast with the surface.

p.21: It is unfortunate that the data base is limited to 314 channels. As you mention in the text it is very likely that IASI-NG will be using channels not in this set.

---

## Author Response (AR1)

Dear Editor,
Please find below our answers to the comments of both referees and of Pasquale Sellitto, how we have modified the text and a marked-up manuscript version. We have expanded the our results and we have tried to discuss their representativity in the text and in the conclusion. At the end, I join the modified version of manuscript.

Yours faithfully,
Nadia Fourrié

**Response to Referee 1**

We would like to thank reviewer 1 for her/his helpful comments and corrections, whic helped to improve the quality of the manuscript. Reviewer comments are reproduced in italic, our anwers are in plain text.

*This paper documents the production of a set of databases designed to simulate IASI and IASI-NG radiances for future use in retrieval and data assimilation studies. It describes in detail how the origin of the "true" atmospheric states used in the calculations and the similarities and differences between the two radiative transfer models employed- RTTOV and 4A.*
*The paper achieves this goal well and (with some minor suggestions outlined below) can be accepted for publication based on that - although the amount of truly new science is limited.*
*The final section presents a somewhat simplistic evaluation of the relative retrieval skill from IASI and IASI-NG. I am assuming it is being presented as an example of the sort of thing that could be done with the database, but I do not think it necessarily adds much to the paper.*
Reviewer 1 is true, the short retrieval study was given as an example of what could be done with the database. According the reviewer2 comments on these results have been added in the conclusion.

*Detailed comments:*
- *p.2 lines 13-14: I don't think you should say that "some channels were unsuited" as the real issue is not the channels themselves but that the information is redundant. Maybe say " a subset of channels is preferred"*
  The proposed change will be included in the text.

- *p.4 line 18: "an adequate refractive index" -> "an appropriate refractive index"*
  This will be changed.

- *p.6 lines 4-6: Is the noise added diagonal or is the fact that noise is correlated between channels because of apodisation allowed for? In fact, apodisation is only briefly mentioned - you should state explicitly the apodisation being used.*
  The noise added to the simulations was diagonal, it is a random noise with a zero mean and a standard deviation with the value of NedT. No error correlation between channels were taken into account. No apodisation has been used here.  We proposed the following explanation : « *Once the instrument data have been simulated, a random gaussian noise using CNES specifications (Fig. 1) for each instrument and IASI-NG configuration was added to the simulated data. These values are valid at 280K and they were converted at the appropriate scene temperature for each wave number and each profiles. Two different noises were used for the IASI-NG simulations according to the two prism materials currently under consideration. Moreover, no correlation between channels was taken into account.* »

- *p.9, lines 18-20: I think the details on how you do not have some cloud flags because of some processing quirk are confusing and not really relevant.*
  We propose to remove these details in the text.

- *p.12, lines 4-6 and Table 5: I do not think that Table 5 is particularly useful as it aggregates the information in Figure 4 while hiding much of the detail. It is also misleading to say there are no differences in the standard deviations when it is clear from Figure 4 that they do indeed exist. So I would remove this table.*
We agree with this proposal to remove table 5. The paragraph will start wit « The mean values (biases) present differences that are displayed in Figure 4 ».

- *Figure 6: Why did you choose to plot this in radiance units when everything else is in brightness temperature? It makes it very difficult to compare with the other plots.*
We thank the reviewer for this point and we have replaced the figure with plots in brightness temperature.

[Figure]

*Figure 6. Comparison of IASI/IASI-NG brightness temperature simulations in the 730-740 $cm_{-1}$ $CO_2$ absorption window against the Earth IR spectrum computed by 4A using a spectral resolution of 0.001 $cm_{-1}$ for one atmospheric description included in the simulation dataset. Upper panel (a) presents the $CO_2$ absorption lines as obtained from GEISA database, the second panel (b) shows the 4A spectrum calculation. Third and four panels present the simulations of IASI (c) and IASI-NG spectra (d). IASI observations are also included in the IASI simulations panel (green dots).*

*The text will be changed accordingly : « IASI-NG signal presents a higher variability giving values going from 231.5 K to 266.5 K for the 239.2 to 262.4 K of the IASI range. »*

- *Section 5: I think you should make it clear from the start that the retrieval discussion is illustrative. The scene selections (totally clear columns), channel selection (limited to IASI 314); background error (arbitrarily set to 2x the NWPSAF supplied one); simple bias correction and simple assumption that IASI-NG noise is exactly 2x IASI are probably not close to how that data will be used in practice.*
We fully agree with this suggestion and we propose to add this sentence at the beginning of section 5 : « *To illustrate the potential gain brought by IASI-NG, a short study using 1D-Var retrievals and a small subset of clear-sky observations over sea is proposed in this section. Indeed theses conditions represent an easier way to deal with infrared observations even if in the future the IASI-NG data will be used with different assumptions (e. g. assimilation over land and/or for cloudy sky…).* »

- *p.15, lines 28-30: Please state explicitly that the 4A calculations are going to be referred to as "truth" for the rest of the section.*
The sentence « These 4A simulations will be considered as the truth in this retrieval study » will be added in the text. »

- *p.17, line 9: Is it really true that the IASI-NG noise is half of IASI noise for all channels?*
We agree that the IASI-NG noise is not exactly half of IASI noise for all the channels as shown in Figure 1 and there are different scenarii for this noise. However the noise tends to be half of the IASI one especially in bands 1 and 2. We propose to reword this sentence :

« As the IASI-NG noise is assumed to be close to half of IASI noise, the values of the associated observation error R matrix were divided by 4 »

- *p.20, lines 9-13: While the channel selection could well be improved, you should be clear that there is a fundamental issue with very low level retrievals in the infrared due to the lack of contrast with the surface.*
  We agree with the reviewer that in addition to the question of the channel sensitivity to the low tropospheric levels, the issue of the lack of contrast with the surface for the retrieval in the low levels of the atmosphere is a key point. We will change the end of the paragraph by « The worst performance of both instruments in the first atmospheric layers is related to the lack of sensitivity of IASI channels at these levels to the first atmospheric layers in the selection used in Météo-France operational system combined with a possible lack of contrast with the surface. A new channel selection for IASI-NG shall be carried out including channels able to improve this lack of sensitivity, and taking into account the IASI-NG bands 3 and 4 thanks to the IASI-NG noise reduction compared to IASI. The retrieval capability of IASI-NG in the low atmospheric levels shall be also studied with respect to the surface contrast. »

- *p.21: It is unfortunate that the data base is limited to 314 channels. As you mention in the text it is very likely that IASI-NG will be using channels not in this set.*
  We have only the 314 channels for the IASI observations but simulations are made of 8461 channels pour IASI, and 16921 channels for IASI-NG. These 314 observed channels are used for comparison with the IASI simulations. We propose to specify this :
  « - **Simulations:** Radiances of IASI (in 8461 channels), 16921 channels for IASI- NG A (Kbr) and IASI-NG B (ZnSe).
  - **Real Observations:** 314 IASI brightness temperatures.»

*Response to Reviewer 2:*

We would like to thank the reviewer 2 for her/his helpful comments and suggestions, which helped to improve the quality of the manuscript. Reviewer comments are reproduced in italic, our responses are in plain text.

*This paper provides a relatively straightforward analysis of the differences in performance between IASI and IASI-NG. This is of interest to the Earth Observation community. However, I think the paper as it stands is not suitable for publication in AMT. This is because I think the authors need to address the representativeness of their results, so that the scientific community can assess how, in general, IASI-NG is an improvement on IASI. I provide details in the specific comments below. The authors should also address the other specific comments.*

The aim of this paper it to promote the database built for IASI/IASI-NG and to encourage the Earth Observation community to use it for its needs (Atmosphere, atmospheric chemistry...). The retrieval study that we propose here is an illustrative study of what can be done with it. We agree that 1681 atmospheric profiles are not enough to demontrate a significant impact of IASI-NG with respect to IASI but it gives a flavour of what can be expected using IASI-NG.

*Specific comments:*
*P. 1*
- *L. 21: Please provide more details than ´nexperimental purposes´z.*
AIRS was launched on board the research Aqua satellite in May 2002. Even if there was only one single copy, this instrument paved the way for the exploitation of the following hyperspectral sounders such as IASI and CrIS. These details were added in the text. «Even if there was only one single copy, this instrument paved the way for the exploitation of the following hyperspectral sounders such as IASI and CrIS. »

*P. 2*

- *L. 1: Check grammar, here and elsewhere: ": : :much of the information is: : :".*
  The change has been made.

- *L. 6: Perhaps provide more details of the "conservative" approach.*
The conservative approach consists in assimilating AIRS data only for clear-sky fields-of-views over sea and with a rather limited number of channels (at most 86 channels). The sentence will be changed in to : « Although this improvement was relatively small, it was encouraging as it was obtained assuming a conservative approach, i.e. with the assimilation of AIRS data only for clear-sky fields-of-views over sea and with a rather limited number of channels (at most 86 channels). »

- *L. 8: Usefulness of the channels for what?*
  The study by Fourrié and Thépaut showed that the AIRS near real time channel selection through different experiments seems very reasonable for NWP applications despite the overall slightly smaller information content versus optimally derived channel selections and that it appears to be robust. We propose to replace this sentence to provide more details with : « The study by Fourrié and Thépaut (2003) showed through different experiments that the AIRS near real time channel selection seems very reasonable for NWP applications despite the overall slightly smaller information content versus optimally derived channel selections and that it appears to be robust.»

- *L. 10: What is the operational mission?*
  IASI in onboard the Metop operational satellite series. The sentence will be changed : I »ASI is an infrared Fourier transform interferometer and is the first instrument of this type to fly as a part of the Metop operational satellite series. »

- *L. 14-15: ": : :Medium-range Weather Forecasts".*
  This has been corrected

- *L. 17: Impact on what parameter? What other data did the Met Office assimilate?*
  The impact was assessed through the UK NWP index. Positive impact on gepotential height, mean surface level pressure and wind forecast depending on the verification areas have been found in the global model. The data from HIRS, AIRS, AMSU-A and MHS sounders were already assimilated in the global model.We propose to modify the text and give more details : « The first Met Office tests about the assimilation of IASI radiances showed a positive impact for the global model forecast (Hilton et al, 2009). Data from infrared sounders (HIRS, AIRS) and microwave instruments (AMSU-A and MHS) were already assimilated. The impact is mainly obtained for gepotential height, mean surface level pressure and wind forecast depending on the verification areas ».

- *L. 19: What large-scale variables?*
  The concerned large scale variables are the temperature and the wind fields and the 500 hPa geopotential height. We suggest to change the sentence with : « They showed a quite good impact on the forecast skills for large-scale variables such as tropospheric temperature, wind fields and 500 hPa geopotential height both in the global ARPEGE model (Courtier et al 1991) and the AROME regional model (Seity, et al , 2011). Precipitation forecasts were also improved in AROME. »

- *L. 28: What other instruments?*
  These instruments consist in for example TES, IMG, GOME or MOPITT. These details will be provided in the text : « IASI evidenced a potential good impact of O3 and CO on air quality forecasts and carries on the long-term chemical records started with other instruments such as the Tropospheric Emission Spectrometer (TES), the Interferometric Monitor Greenhouse gases instrument (IMG),the Global Ozone Monitoring Experiment (GOME-2) and Measurement of Pollution in the Troposphere (MOPITT) (Dufour etal, 2012, Hilton et al 2012) »

- *L. 33: I suggest you replace "race" with "effort".*

*This was done.*

**P.3**

- *L. 1: I would suggest you use a word different from "figures".*
  The sentence will be changed to « The former of these two instruments covers the same spectral range as IASI with a noise reduction of at least a factor two and a twice higher spectral resolution. »

- *L. 3 (and elsewhere in this paragraph): OSSEs.*
  The changes have been made.

- *L. 8: Perhaps write: ": : :typically mimic: : :".*
  The modification is accepted.

- *L. 11-13: Explain the advantages of using two different radiative transfer models.*
  4A OP is used by CNES for the simulation of IASI observations in the operations and RTTOV is widely used in the NWP community. Having two different radiative transfer model allows to carry out observation impact study with simulated observations produced by a radiative transfer model different from the one which will be used in the retrieval study. This is commonly done in the field of OSSEs where this difference of RT adds a more realistic error to the radiance data. We suggest to add the followin sentence : « … 4A simulations could be used. This use of a different RT for the retrieval adds a more realistic error to the radiance data. »

- *L. 17: last -> latter.*
  The change has been made.

- *L. 34: Larger noise than what?*
  The IASI noise in band 3 is larger than the ones of the other bands. This was specified in the text. «  It is not used at Météo-France because of its larger noise compared to the two other bands. »

**P. 4**

- *L. 6: If you use British spelling, it should be "programme".*
  The word has been corrected

- *L. 17: What is an adequate refractive index?*
  *As suggested by reviewer 1, the « adequate refractive index » has been replaced with « an appropriate refractive index ».?*

- *L. 17-18: I suggest you rephrase "The noise requirements: : : two."*
  *We propose the following sentence : « One of the requirement of IASI-NG is to have a noise at least divided by a factor two compared to the IASI one. »*

**P. 5**

- *Table 1: There seems to be a missing entry (NedT for IASI).*
  NedT for IASI is varying across the band and is given in Fig. 1 for IASI. We have changed the Table as following :

| Main features | IASI | IASI-NG |
|---|---|---|
| Pixels in field of view | 4 | 16 |
| Channels | 8461 | 16921 |
| Radiometric resolution (NedT) | see Fig. 1 | IASI/2 |
| Spectral sampling | $0.25\ cm^{-1}$ @L1C | IASI/2 |
| Abs. Radiometric Calibration | <0.25 K @280K | IASI/2 |
| Spectral bands | 3 | 4 |

**Table 1.** main features of IASI and IASI-NG sounders

- *Sect. 3: The database uses one day per season. Could you discuss if this is enough to be representative of seasonal conditions?*
  We agree that with only 4 days of IASI data the representativity of our dataset may be quite limited depending on specific applications. However it represents more than 5 millions of observation points over the globe and we have tried to cover a maximum of situations. Each day represents a volume of 1 Terabytes.
  A discussion on the possible weak representativity has been introduced : « Even though the dataset represents more than 5 millions of IASI observation points (5242448) over the globe, it is likely that these 4 days do not cover all the possible meteorological situations. However it offers the possibility to have a variety of atmospheric profiles covering the whole Earth, with daytime/night-time and sea/land conditions. »

*P. 6*
- *L. 5: Provide more details of how you apply the noise. There are four NedT curves in Fig. 1. For each material and each instrument, the noise added to the simulations for the observations corresponds to the ones given in Fig1. No correlation between channels was taken into account. These values are valid at 280K and were converted at the appropriate scene temperature for each wave number and each profile.*
  *We propose to add this explanation in the paper : « Once the instrument data have been simulated, a random gaussian noise using CNES specifications (Fig. 1) for each instrument and IASI-NG configuration was added to the simulated data. These values are valid at 280K and were converted at the appropriate scene temperature for each wave number and each profiles. Two different noises were used for the IASI-NG simulations according to the two prism materials currently under consideration. Moreover, no correlation between channels was taken into account. »*

- *L. 11: You use the MACC data for each of the four dates you mention previously in the paper?*
  Yes this is true. This has been specified in the paper. « In order to achieve this goal, the vertical profiles from a selection of atmospheric constituents in the different IR absorption bands measured by both instruments were extracted for each date from global analyses provided by the Monitoring Atmospheric Composition and Climate 5 (MACC) project of the Copernicus programme »

*P. 7*
- *L. 29: Provide examples of this use of RTTOV.*
  RTTOV is used for radiance data assimilation at Météo-France, Met Office and ECMWF for example. The atmospheric retrievals can be obtained through the use of the 1D-Var provided by the EUMETSAT NWP SAF. Simulated satellite imagery are used by forecasters for nowcasting or to compare the forecast with the satellite observations.
  We propose to change the text by « RTTOV (Matricardi et al, 2004 ; Hocking et al, 2015) is a fast RT model for passive visible, infrared (IR) and microwave (MW) satellite-borne sensors and is used in various applications. Radiance data assimilation make use of RTTOV

at Météo-France, Met Office and ECMWF for example. Simulated satellite imagery is used by forecasters to compare the forecast outputs with the satellite radiances for nowcasting. Moreover, atmospheric retrieval can be obtained through the use of the 1D-Var provided by the EUMETSAT NWP Satellite Application Facility. »

*P.8*

- *L. 22-23: I do not understand this phrase.*
  The paragraph has been rewritten as follows : « RTTOV simulations were run over 60 levels with the latest coefficient files.The version 11 of RTTOV could be only supplied with 6 atmospheric constituents as inputs (compared with the 43 species possible with 4A) and only 5 vertical profiles of atmospheric constituents were provided to the model: H2O, CO2, O3, CO, and CH4. The 6th possible atmospheric species, but not used in our case, is N2O. »

*P. 9*

- *L.7: Perhaps the authors could provide a reference for the statement about uncertainties of cloud radiative properties.*
  The text was incorrect, We meant « Although the AVHRR cloud cover value corresponding to each IASI pixel and cloud vertical profiles have been included in the database, we have chosen not to consider cloudy conditions because of computational cost of these kind of simulations and also due to the large uncertainties in cloud radiative properties modeling. »

- *L. 15: ": : :five latitude belts: : :".*
  The proposed change was made.

*P. 10*

- *L. 7: What is the point of Fig. 3?*
  Figure 3 aims at showing an example of a large part of all the information contained in the database.

*P. 12*

- *L. 9: Introduce acronyms like ppmv. Grown up -> increased*
  Suggested modifications were made in the text.

- *L. 10: Clumsy use of "outdating". Reword.*
  The sentence has been replaced with « … approximately, making this RTTOV profile training set possibly obsolete. »

*P.14*

- *L. 5-6: Clumsy construction; please rephrase.*
- We propose the following clarification: « As it has been found that IASI reconstructed radiances from a principal component compression do not exhibit this artefact Hultberg(2010), an additional comparison between the simulated spectra and reconstructed radiances was carried out (Figure not shown). »

*P. 19*

- *Fig. 8: Indicate the meaning of the colours in the plots.*
  There is no meaning of the colours in the plots. They only contribute to show the individual contribution of each channel.

*P. 20*

- *L. 5: Avoid subjective words like "striking".*
  Striking has been replaced with « different »

- *L. 17: This explanation is not very clear to me. Please clarify.*
  We propose the following clarification : « These smaller error reductions are due to the lack of sensitivity of IASI channels to this atmospheric region combined to a weak background error in the humidity associated to these levels compared to the other regions ones preventing observations to add more information in the 1D-Var. »

- *L. 13+: As I see it, this paper shows that IASI-NG performs better than IASI. Is this to be expected? I presume that the value of the paper is that you quantify this improvement. How representative is this improvement? I would suggest that you discuss these points in the conclusions section.*

  As explained above, the aim of this paper was to present the IASI/IASI-NG database is available to the Earth Observation community. The impact illustrated in this paper is rather to demonstrate that these data are usable. Please find below the modifications in red brought in the last paragraph of the conclusion :

  Two kinds of validations have been presented :  the first one consists in the evaluation of the IASI simulations and the second one is an evaluation of the gain brought by IASI-NG with respect to the IASI one with 1D-Var experiment. For one single orbit, all clear cases over sea and during night-time, the RTTOV and 4A simulations have been compared to the measured IASI 8,461 channels. Similar results were found for the two models. On one hand by using the sea surface emissivity ISEM model, RTTOV has slightly better results than 4A. On the other hand, land-surface emissivities from 4A are better because of using UW emissivities for the 2013 instead of 2007 like RTTOV. RTTOV appears to better represent the water continuum providing a lower bias value in IASI band 2. By simply considering the noise reduction of IASI-NG the improvement on brightness temperature error reduction mainly varies between 5 and 15 percent compared to IASI.

  With a small subset of atmospheric profiles from the database and an 1D-Var framework, the impact of the current IASI channel selection been evaluated for IASI and IASI-NG configurations over sea and for clear sky conditions.   With channels located in the same wave numbers, retrieval experiments showed that the retrieval with IASI-NG improves temperature profiles along all the atmosphere with a maximum in the troposphere. This improvement may reach 10 % in the Tropics at 400 hPa and is about 5 % in the Mid-latitudes and in polar regions. The improvement obtained for tropospheric humidity is of the same order (5%).

   A reduced sensitivity in the low troposphere is confirmed for IASI.  This result agrees with the study by Sellitto et al. (2013) who produced tropospheric ozone pseudo-observations based on this noise configurations. They showed a clear improvement of low tropospheric ozone pseudo-observations compared to the IASI ones and the potential to separate lower from upper tropospheric ozone information.

  Additional work is thus required to check if IASI-NG will be able to better probe the atmosphere at these levels  For this purpose, a new channel selection needs to be defined, which will be undertaken in a following study.

  These encouraging results of the IASI-NG impact provided by this study should be confirmed with a more comprehensive atmospheric dataset and a closer context of NWP operations. IASI-NG is dedicated to multiple applications such as NWP, atmospheric chemistry and air quality. The potential of synergy between instruments from EPS second Generation should be studied for the various application such as ozone pollution as proposed by Costantino et al. (2017).

Response to comment by Pasquale Sellitto

*Please note that the following paper discusses the added-value of one possible configuration of IASI-NG in the characterisation of the lower troposphere in terms of the ozone concentration: Sellitto, P., Dufour, G., Eremenko, M., Cuesta, J., Dauphin, P., Forêt, G., Gaubert, B., Beekmann, M., Peuch, V.-H., and Flaud, J.-M.: Analysis of the potential of one possible instrumental configuration of the next generation of IASI  instruments to monitor lower tropospheric ozone, Atmos. Meas. Tech., 6, 621-635,  https://doi.org/10.5194/amt-6-621-2013, 2013. Even if your paper does not directly address the topic of air quality, I think that citing this work would be useful*

*when discussing your results (and IASI-NG expected added-value), see e.g. this sentence in your conclusions: "A reduced sensitivity in the low troposphere is confirmed for IASI and, additional work is required to check if IASI-NG will be able to better probe the atmosphere at these levels."*

We would like to thank Pasquale Sellitto for his helpful suggestion to include this reference to the improvement brought by IASI-NG for the determination of ozone concentration in the low troposphere. The text will be changed as following : « A reduced sensitivity in the low troposphere is confirmed for IASI. This result agree with the study by Sellito et al 2014 who produced tropospheric ozone pseudo-observations based on this noise configurations. They showed a clear improvement of low tropospheric ozone pseudo-observations compared to the IASI ones and the potential to separate lower from upper tropospheric ozone information. Additionall work is thus required to check if IASI-NG will be able to better probe the atmosphere at these levels ».

*In addition, I also suggest to open your discussion to possible multi-spectral synergies, with reference to the following paper: "Costantino, L., Cuesta, J., Emili, E., Coman, A., Foret, G., Dufour, G., Eremenko, M., Chailleux, Y., Beekmann, M., and Flaud, J.-M.: Potential of multispectral synergism for observing ozone pollution by combining IASING and UVNS measurements from the EPS-SG satellite, Atmos. Meas. Tech., 10, 1281-1298, https://doi.org/10.5194/amt-10-1281-2017, 2017." I suggest adding these two references to put your very useful work in a slightly wider context.*

We can open the discussion if the editor agrees with this suggestion. We will add this short paragraph at the end of the conclusions

[revised manuscript text omitted]

---

## Author Response (AR2)

Dear Editor,

Thank you very much for your corrections. I have added the following sentence to the caption of Fig. 8 :"The various colours correspond to the different channels to show their individual contribution". In addition I changed the two last sentences according to your suggestion.

Your faithfully,

Dr Nadia Fourrié